# Attentional efficiency does not explain the mental state × domain effect

**Joseph Sweetman** *, **George A. Newman**

Department of Psychology, College of Life and Environmental Sciences, University of Exeter, Exeter, Devon, United Kingdom

* j.p.sweetman@exeter.ac.uk

## Abstract

The reduced importance of intent when judging purity (vs. harm) violations is some of the strongest evidence for distinct moral modules or systems: moral pluralism. However, research has indicated that some supposed differences between purity and harm moral domains are due to the relative weirdness of purity vignettes. This weirdness might lead to a failure to attend to or correctly process relevant mental state information. Such attentional failures could offer an alternative explanation (to separate moral systems) for the reduced exculpatory value of innocent intentions for purity violations. We tested if the different role of intent in each domain was moderated by individual differences in attentional efficiency, as measured by the Attention Network Task. If attentional efficiency explains the reduced exculpatory value of innocent intentions in purity (vs. harm) violations, then we would expect those high (vs. low) in attentional efficiency not to show the reduced exculpatory effect of innocent intentions in the purity (vs. harm) domain. Consistent with moral pluralism, results revealed no such moderation. Findings are discussed in relation to various ways of testing domain-general and domain-specific accounts of the mental state × domain effect, so that we might better understand the architecture of our moral minds.

## Introduction

Intentionally slapping a sibling in the face is morally wrong, whereas performing the exact same action accidentally, as you go to high five them, is not. In this case one's intent, or lack thereof, distinguishes the moral status of the two acts. However, this distinction does not hold, or is much weaker at least, for intentional vs. accidental incest (e.g., sex between siblings separated at birth). Moral pluralism explains these facts by positing that our moral judgments of cases involving harm and "purity" (e.g., incest, drinking urine) are underpinned by separate, domain-specific moral modules or systems. We test an alternative explanation. Namely, that differences in the exculpatory effect of innocent intentions are due to attentional failures brought on by the relative "weirdness" of purity violations.

### Moral pluralism and the role of intent across moral domains

Moral pluralism holds that our moral capacity is underpinned by separate, domain-specific moral modules or systems. For example, moral foundations theory (MFT) posits that humans

**Data Availability Statement:** The data underlying the results presented in the study are available at https://osf.io/8k2hj/

**Funding:** The author(s) received no specific funding for this work

**Competing interests:** The authors have declared that no competing interests exist

have distinct mental modules that each process information for specific moral domains and related actions (e.g., violations): harm–triggered by suffering, distress, or neediness (e.g., assault), purity–triggered by bodily fluids, taboo diets and sexual practices (e.g., incest), fairness–triggered by failure to cooperate and share resources (e.g., cheating), loyalty–triggered by undermining coalitions and intergroup competition (e.g., betrayal), and authority–triggered by undermining intragroup status hierarchies (e.g., disobedience) [1,2]. MFT claims that each of these mental modules has evolved through natural selection to enable fast and efficient responses to recurrent adaptive problems. Some evolutionary biologists have suggested that while it is tempting to provide adaptationist "stories" for aspects of human cognition, it may prove impossible to empirically test these as scientific hypotheses [3]. Regardless of our ability to answer how human moral cognition evolved, understanding whether it is constituted by a single system or separate moral systems (i.e., moral pluralism) is a fundamental question for the science of morality. Indeed, correctly delineating and decomposing cognitive phenomenon is a standard way of developing theoretical explanation in the psychological and cognitive (neuro)sciences [4].

Key evidence for moral pluralism comes from Young and Saxe (2011) who showed that the exculpatory effect of innocent intentions was significantly reduced for purity compared to harm violations [5]. For example, accidentally poisoning a dinner guest with an undisclosed peanut allergy was judged less morally wrong than accidentally committing incest with a long-lost sibling (Experiments 1–3). However, intentional harm (e.g., poisoning) was either judged worse than or the same as intentional purity violations. Put differently, the simple main effect of mental state (intentional vs. accidental) was stronger for violations in the harm (vs. purity) domain. Subsequently, converging evidence for this mental state × domain effect has been found both in fMRI and cross-cultural work [6,7]. Specifically, imaging data shows that the right temporoparietal junction (RTPJ; an area recruited for mental state reasoning or theory of mind) was preferentially recruited for processing harmful vs impure acts. Multivoxel pattern analysis revealed that RTPJ distinguished the mental state of the agent for harm, but not purity, violations [6]. Findings across a number of traditional small-scale societies (e.g., hunter-gatherer and pastoralist) also provide support the mental state × domain effect [7]. Furthermore, the original finding has been supported in an independent, pre-registered, close replication [8]. Taken together, these findings seem to offer some evidence that moral cognition for cases of harm and purity violations may be underpinned by separate, domain-specific moral systems.

## Moral systems or attentional failures

Critics of moral pluralism claim that the above evidence for separate, domain-specific moral systems is better explained as a feature of how psychologists operationalise moral domains in their experiments [9]. Moral domain is not the only difference between acts of poisoning (i.e., harm) and incest (i.e., purity). Gray & Keeney found that purity violations (e.g., incest, drinking urine, etc.) are both "weirder" (e.g., strange, unexpected) and less severe (e.g., serious) than cases of harm (e.g., assault) employed in this literature [9]. The authors claim that it is this scenario sampling bias, not the type of moral domain, that is responsible for differing patterns of moral judgement. Using scenarios that were matched across weirdness and severity they found that the weirdness of the scenario was more predictive of character evaluations than was moral domain. This offers an alternative explanation (to moral pluralism) of earlier work that had suggested that purity (vs. harm) violations lead to less severe ratings of action but harsher judgments of moral character [10].

Returning to the role of intent across moral domain, it is also possible that differences in the scenarios employed (other than moral domain) may explain the mental state x domain interaction. Moreover, it might not be the weirdness of scenarios, per se, that is the proximate

cause of these effects. Rather, the relative weirdness of purity (vs. harm) scenarios may be a more distal factor that impacts the computations underlying moral cognition via more general cognitive processes–namely, attentional efficiency.

Huebner, Dwyer, & Hauser, 2009 offer a similar perspective when considering the role of emotion in moral psychology [11]. Specifically, the authors argued that emotion could impact moral judgments through its impact on attentional processes. Analogously, the weirdness of purity (vs. harm) scenarios may compete with standard moral cognition for attentional resources, leading to a failure to properly process mental state information in these cases [8]. Indeed, research suggests that unexpected and surprising stimuli, which we argue purity violations like incest and drinking urine are clear cases of, compete for attentional resources and can lead to "surprise-induced blindness" in attentional processes [12]. This may explain the reduced exculpatory effect of innocent intentions in the purity (vs. harm) domain without having to evoke separate, domain-specific moral systems (see Fig 1 for a graphical depiction of these alternative explanations).

In short, if attention failures explain the reduced exculpatory value of innocent intentions in purity (vs. harm) violations, then we would expect individual differences in attentional processing efficiency to moderate the mental state x domain interaction. Specifically, those high (vs. low) in attentional efficiency should not show the reduced exculpatory value of innocent intentions in the purity (vs. harm) domain. Alternatively, if modular, domain-specific information processing explains the reduced exculpatory value of innocent intentions across domain, then we would expect the mental state × domain interaction to be the same, or increase in magnitude, in those high (vs. low) in attentional efficiency (see below for further details of these hypotheses).

## The present study

In this experiment we used the attentional network task (ANT) [13] as a measure of attentional processing efficiency in order to test our hypotheses. The ANT (see section 2.1.5.1) evaluates three functions of attention: alerting, orienting, and executive control. Alerting is thought of as achieving and maintaining a state of high readiness to receive information. Orienting refers to the direction of attention towards specific sensory information. Finally, executive control is involved with mental resource recruitment to resolve conflicts and act contrary to automatic biases, expectations, or habits [14].

If attentional failure is responsible for the reduced exculpatory value of innocent intentions in purity (vs. harm) violations, then individual differences in attentional network efficiency should moderate this effect. The weirdness of the purity violations (e.g., incest, drinking urine) could plausibly cause failures in any one of the three attentional networks. Specifically, the weirdness of purity dilemmas could make it more difficult to integrate mental state information into moral cognition, something that may depend crucially on attentional control [15–18]. As such, we might expect executive control ANT scores to moderate the mental state × domain interaction. Alternatively, the weirdness of purity violations may influence how ready individuals are to receive information about other aspects of the scenario (e.g., the action, outcome, intent) meaning that we would expect the alerting ANT score to moderate the effect. Finally, the weirdness of purity violations may compete with standard moral cognition for attentional resources, leading to the failure to properly encode or process mental state information in these cases (for an analogous argument, see [11]). If this is the case, then the orienting ANT score should moderate the mental state × domain effect.

In all cases, if the reduced exculpatory value of innocent intentions in purity (vs. harm) violations is due to attentional failure, then the mental state × domain interaction should be

**Explanation 1**. Moral pluralism

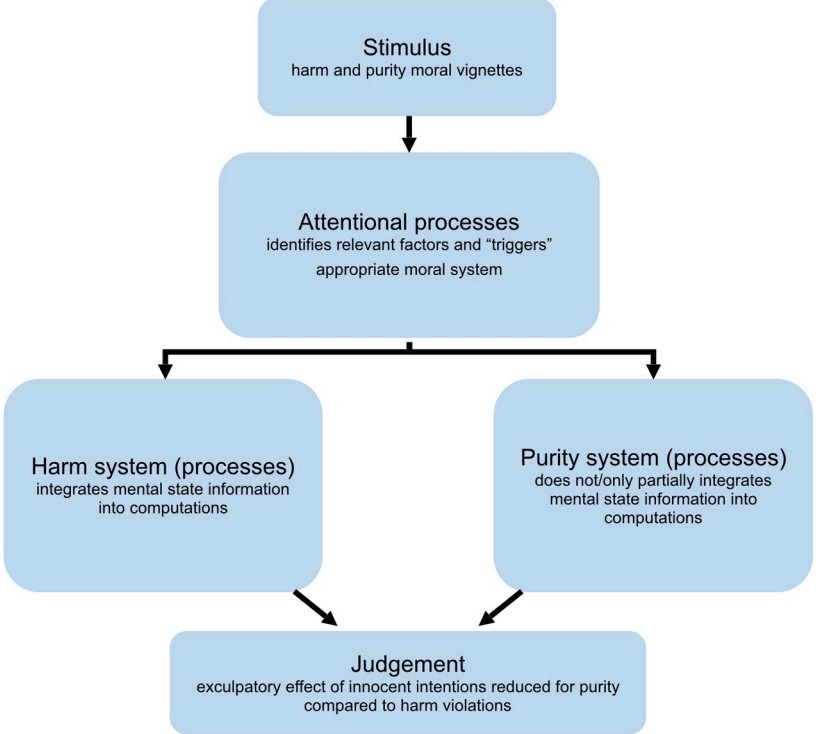

**Explanation 2**. Attentional failure

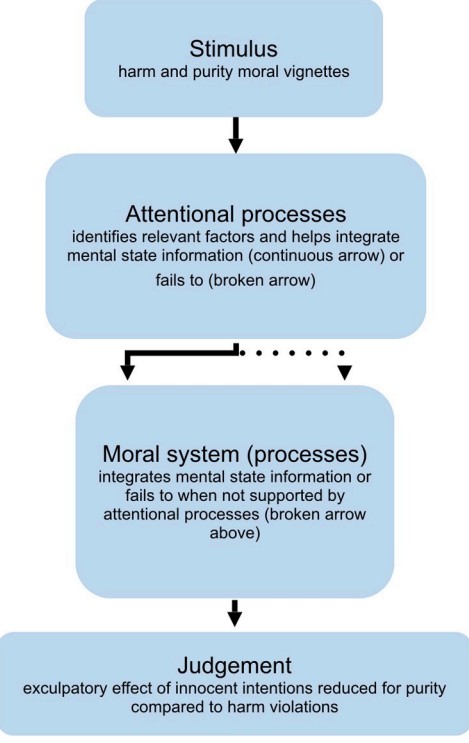

**Fig 1. Explanations of the mental state × domain effect.** Explanation 1. Moral pluralism: modular, domain-specific information processing explains the reduced exculpatory value of innocent intentions across domain. Explanation 2. Attentional failure: the weirdness of purity vignettes interferes with attentional processes that help to integrate mental state information into moral cognition, leading to the reduced exculpatory value of innocent intentions in the purity domain.

attenuated for individuals that have more efficient attentional processes (i.e., those with lower ANT scores). Statistically, the mental state × domain interaction should be moderated by individual differences in ANT scores (i.e., attentional efficiency).

## Materials and methods

### Pilot

Due to concerns about the functionality of the experimental code a pilot study of 20 participants was run before (07/01/2019) the main data collection. As no issues were encountered and the methodology was not changed this pilot data was included in final data analysis.

### Participants and pre-registration

570 participants (including 20 from the pilot) were recruited through the online platform Prolific (prolific.ac) of which 545 successfully returned data files. Due to a coding error, participant gender and age were not recorded. We had no reason to expect any meaningful role of age or gender in explaining the mental state × domain effect. Before participation in the study, participants gave their informed consent. The materials used in the study had previously be granted ethical approval from the Psychology ethics committee at the University of Exeter (eCLESPsy001180 v3.4). We estimated the completion time of the study at approximately 14 minutes. Participants were therefore paid £1.17 to complete the survey at an hourly rate of £5.02. The study was preregistered with the OSF on 07/01/2019 after the analysis of the pilot study but before collection of the remaining data and the main analysis (see details at https://osf.io/8k2hj/).

### Design and procedure

The experiment was a 2 (mental state: intentional vs. accidental) x 2 (domain: harm vs. purity) within-participants design with ANT scores as continuous moderators. All participants completed the ANT and the moral judgements in two separate sections of the experiment with the order of these tasks counterbalanced. Within the moral judgment section, participants were presented with the four conditions in a random order. Power calculations (GPower 3.1) showed that 68 participants would provide 95% power to detect a small to moderate effect size, $\eta_p^2 = .05$. This is based on a within-between (ANT: high vs. low) design with a correlation of $r = .2$ between the within-participants factors (mental state and domain). However, recent commentary has suggested that when looking for interactions that "knock out" or attenuate a given effect the sample size should be multiplied by a factor of four to sixteen, respectively [19,20]. However, due to limited resources we could only afford to increase the sample by a factor of eight: giving us a target N of 544.

### Materials and measures

The research materials consisted of an online experiment developed using the Inquisit 4 environment (details of all scripts and materials at https://osf.io/8k2hj/).

**ANT.** The ANT is a well-validated measure of attentional processes both in the lab and online [14,21,22]. The task evaluates three functions of attention: alerting, orienting, and executive control (responsible for detecting and resolving conflict). It does this by requiring a participant to indicate the direction of an arrow, that appears on screen, as fast as possible and recording the accuracy and reaction time. The arrow is flanked by distractor arrows that are either pointing in the same direction as the central one (congruent condition) or in the opposite direction (incongruent). Additionally, the appearance of the arrows is sometimes preceded by a cue that tells the participant that the stimuli are about to appear. The cue sometimes includes location information, so that the participant knows whether to expect the arrows above or below a central fixation point (see Fig 2). By comparing the reaction times of all of the various conditions the three attentional network scores can be calculated: Alerting = RT for no-cue–RT for double-cue, Orienting = RT for center-cue–RT for spatial-cue, and Executive control = RT for incongruent–RT for congruent stimuli.

We used the "Centre for Research on Safe Driving Attentional Network Task (CRSD-ANT)–Arrows" version of the test [22] taken from the Inquisit test library. This is a short (10-minute) online version of the task that has been shown to produce results that are highly correlated with traditional longer versions [22]. The ANT consisted of 1 block of 32 practice trials with feedback, followed by 2 blocks of 64 trials each. Each block was separated by a rest break. Participants had to manually indicate when they were ready to begin the next block. Ordering of trial types within blocks is random. The main measures were the three network scores. Greater network scores indicate *less* attentional network efficiency.

**Moral judgement..**   All participants made a moral judgement of the four scenarios with the order of presentation randomized. The scenarios were written in third-person such that participants were judging the actions of a gender-neutral agent (Sam). The scenarios were taken from Experiment 1B of Young and Saxe's (2011) paper [5]. The harm violation involved intentionally (or accidentally) poisoning a cousin, while the purity violation involved intentionally (or accidentally) committing incest with a long-lost sibling (see materials at ***https://osf. io/8k2hj/***). Participants judged the moral wrongness of the action described in the scenarios on a 7-point scale, anchored at "not at all morally wrong" (1) to "very morally wrong" (7).

## Results

We used R to perform a linear mixed effects analysis of the relationship between domain, mental state, attentional network efficiency and moral judgment. As fixed effects, we entered domain, mental state, and ANT scores (mean-centred), with all two- and three-way interaction terms, into the model. We specified three separate models to separately test whether each attentional network score moderated the mental state × domain interaction. We specified a random intercept for participants in each model (see Table 1 for details of the models). Visual inspection of residual plots did not reveal any obvious deviations from normality or homoscedasticity. Inspection of influence statistics (leverage and Cook's distance) did not reveal any obvious influential cases (e.g., Cook's < .2). We also specified Bayesian versions of the same models, with uninformative priors. All data, scripts, Supplementary Figs and Descriptives are available at https://osf.io/8k2hj/.

### Moderation of mental state × domain interaction by attentional network

As shown in Fig 3, analyses revealed little support for moderation of the mental state × domain effect, with the magnitude of the effect remaining remarkably similar at high (vs low) attentional efficiency across all networks. Specifically, we found little support for the three-way interaction between mental state × domain × alerting network score, $b$ = .003, 95% CI [-.001,

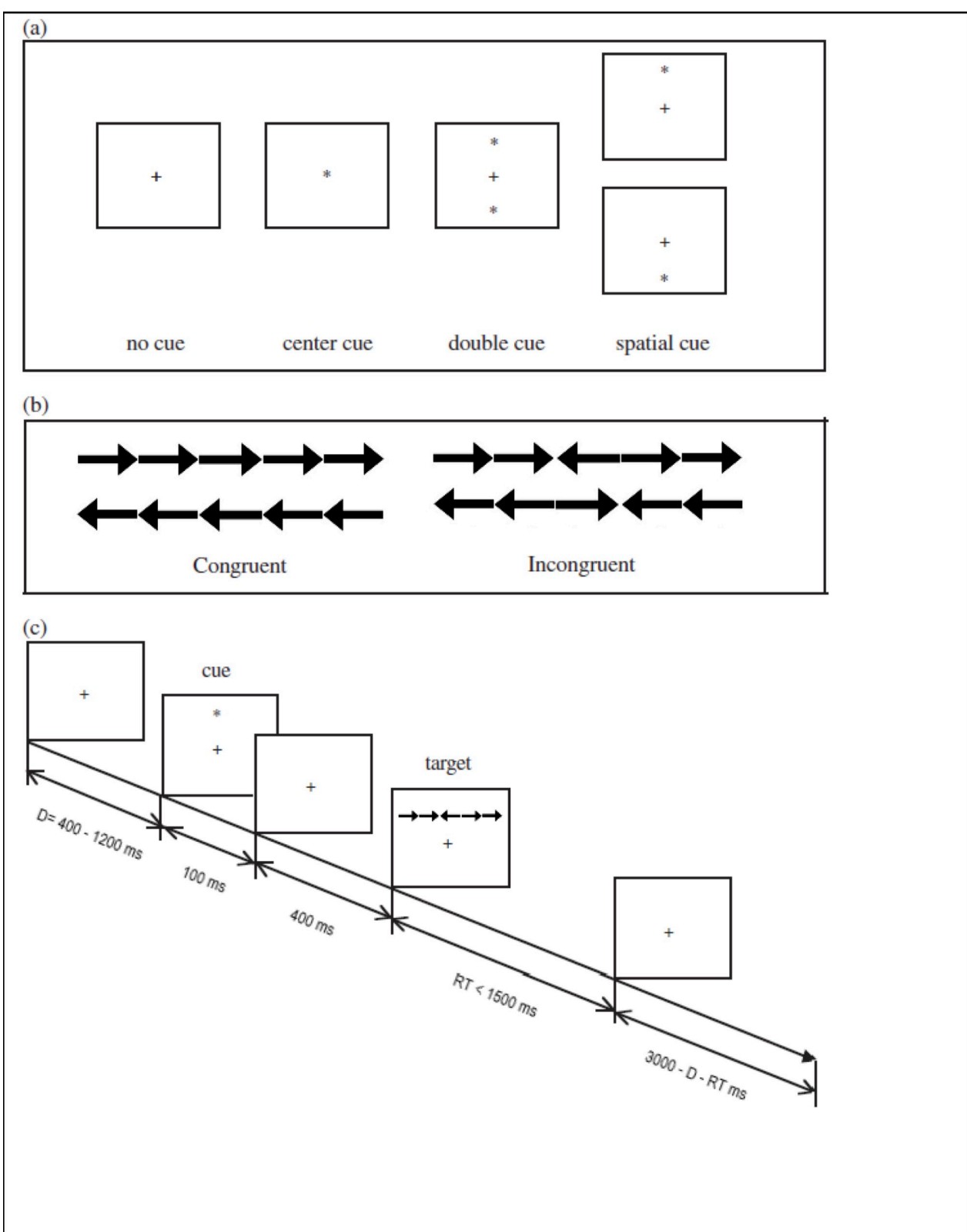

**Fig 2. Experimental procedure.** (a) The four cue conditions; (b) The four stimuli used in the present experiment; and (c) An example of the procedure.

**Table 1. Model estimates for the mental state × domain × attentional network linear mixed effects models.**

| | Moral judgment | | |
| | Attentional network model | | |
| | (1) | (2) | (3) |
| Constant | 4.124*** | 4.124*** | 4.124*** |
| | (4.041, 4.206) | (4.041, 4.206) | (4.041, 4.206) |
| Domain | -0.267*** | -0.267*** | -0.267*** |
| | (-0.408, -0.126) | (-0.408, -0.127) | (-0.409, -0.126) |
| Mental state | -3.356*** | -3.356*** | -3.356*** |
| | (-3.497, -3.215) | (-3.497, -3.215) | (-3.497, -3.215) |
| Alerting | 0.001 | | |
| | (-0.001, 0.002) | | |
| Orienting | | -0.0001 | |
| | | (-0.002, 0.001) | |
| Executive | | | 0.0005 |
| | | | (-0.001, 0.001) |
| Domain:Mental state | 1.561*** | 1.561*** | 1.561*** |
| | (1.279, 1.843) | (1.279, 1.843) | (1.278, 1.843) |
| Domain:Alerting | -0.001 | | |
| | (-0.003, 0.002) | | |
| Mental state:Alerting | -0.001 | | |
| | (-0.003, 0.001) | | |
| Domain:Mental state:Alerting | 0.003 | | |
| | (-0.001, 0.008) | | |
| Domain:Orienting | | -0.0003 | |
| | | (-0.003, 0.002) | |
| Mental state:Orienting | | 0.003* | |
| | | (0.0001, 0.005) | |
| Domain:Mental state:Orienting | | 0.002 | |
| | | (-0.003, 0.008) | |
| Domain:Executive | | | 0.001 |
| | | | (-0.001, 0.003) |
| Mental state:Executive | | | -0.0001 |
| | | | (-0.002, 0.002) |
| Domain:Mental state:Executive | | | 0.001 |
| | | | (-0.002, 0.005) |
| Observations | 2176 | 2176 | 2176 |
| Log Likelihood | -4325.497 | -4324.623 | -4327.336 |
| Akaike Inf. Crit. | 8670.994 | 8669.245 | 8674.672 |
| Bayesian Inf. Crit. | 8727.846 | 8726.098 | 8731.525 |

(1) mental state × domain × alerting network; (2) mental state × domain × orienting network; (3) mental state × domain × executive control network. Factors were deviation coded–domain: -.5 harm, .5 purity; mental state: -.5 intentional, .5 accidental. ANT scores were mean-centered.

*p < .05;

**p < .01;

***p < 0.001. 95% confidence intervals are present within brackets.

.008], $p = .121$, $R_\beta^2 = .001$, 95% CI [.000, .005]. Put differently, alerting network scores did not significantly moderate the mental state × domain effect. The reduced exculpatory value of innocent intentions in purity (vs. harm) violations can be seen at low (alerting score mean +1

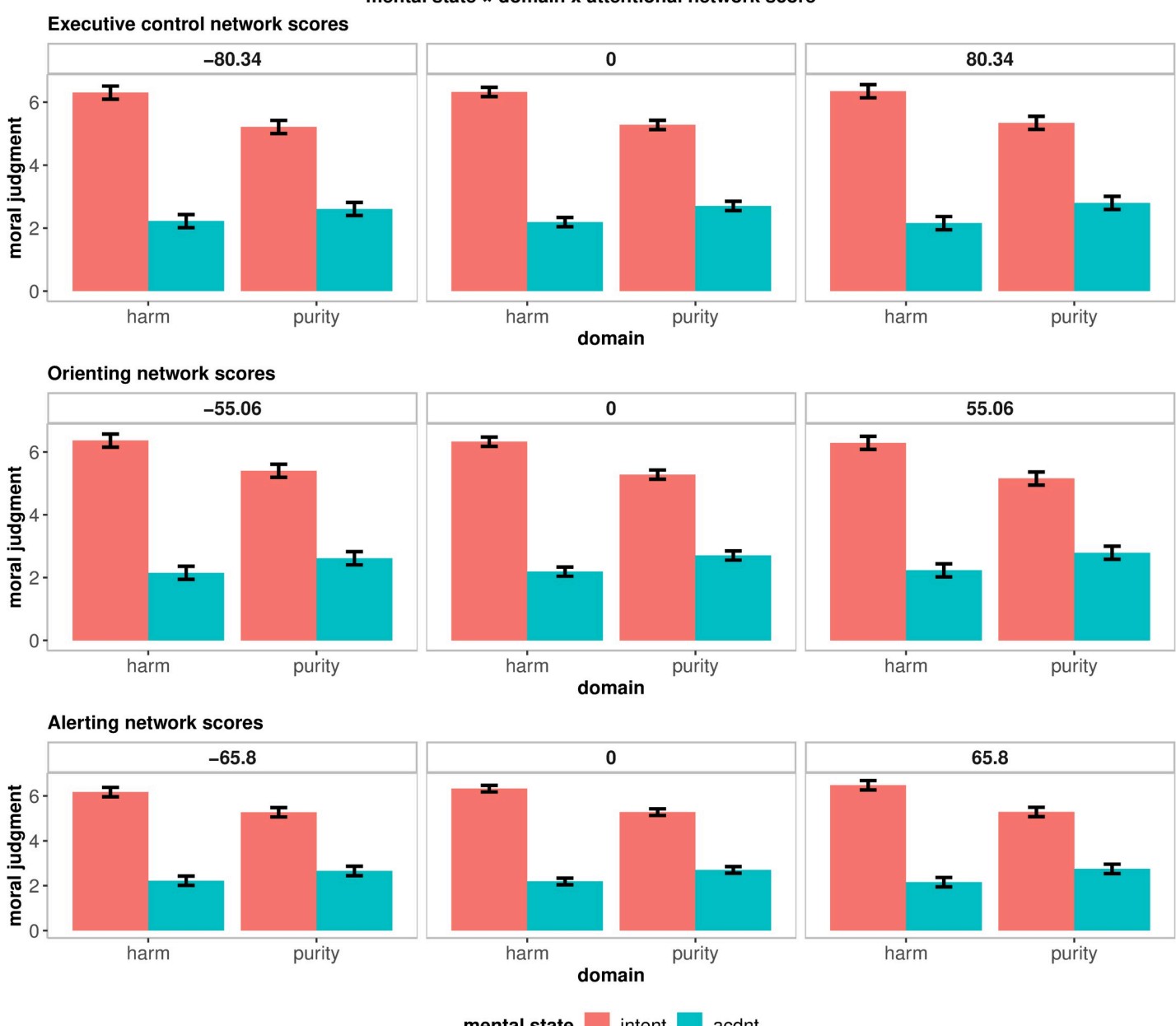

**Fig 3. Mental state × domain as a function of attentional network score.** Alerting efficiency (bottom panel), orienting efficiency (middle panel), and executive control efficiency (top panel). Error bars reflect 95% CIs.

SD) and high (alerting score mean -1 SD) levels of alerting efficiency (see bottom panel in Fig 3). Similarly, the mental state × domain effect was not significantly moderated by orienting network scores, $b$ = .002, 95% CI [-.003, .008], $p$ = .351, $R_\beta^2$ = .000, 95% CI [.000, .004]. The two-way interaction pattern remains at low (orienting score mean +1 SD) and high (orienting score mean -1 SD) levels of orienting efficiency (see middle panel in Fig 3). Furthermore, analyses revealed little support for the mental state × domain × executive control three-way interaction, $b$ = .001, 95% CI [-.002, .005], $p$ = .551, $R_\beta^2$ = .000, 95% CI [.000, .003]. In other words, executive control network scores did not significantly moderate the reduced exculpatory value

**Table 2. Model estimates for the mental state × domain × attentional network Bayesian linear mixed effects models.**

| | Moral judgment | | |
| --- | --- | --- | --- |
| | **Attentional network model** | | |
| | **(1)** | **(2)** | **(3)** |
| Constant | 4.124 | 4.124 | 4.124 |
| | (4.041 – 4.205) | (4.041 – 4.206) | (4.042 – 4.206) |
| Domain | -0.267 | -0.268 | -0.268 |
| | (-0.410 – -0.123) | (-0.410 – -0.124) | (-0.408 – -0.126) |
| Mental state | -3.355 | -3.356 | -3.355 |
| | (-3.497 – -3.213) | (-3.498 – -3.214) | (-3.495 – -3.215) |
| Alerting | 0.001 | | |
| | (-0.001 – 0.002) | | |
| Orienting | | -0.000 | |
| | | (-0.002 – 0.001) | |
| Executive | | | 0.000 |
| | | | (-0.001 – 0.001) |
| Domain:Mental state | 1.561 | 1.561 | 1.560 |
| | (1.277 – 1.846) | (1.277 – 1.843) | (1.279 – 1.840) |
| Domain:Alerting | -0.001 | | |
| | (-0.003 – 0.002) | | |
| Mental state:Alerting | -0.001 | | |
| | (-0.003 – 0.001) | | |
| Domain:Mental state:Alerting | 0.003 | | |
| | (-0.001 – 0.008) | | |
| Domain:Orienting | | -0.000 | |
| | | (-0.003 – 0.002) | |
| Mental state:Orienting | | 0.003 | |
| | | (0.000 – 0.005) | |
| Domain:Mental state:Orienting | | 0.002 | |
| | | (-0.003 – 0.008) | |
| Domain:Executive | | | 0.001 |
| | | | (-0.001 – 0.003) |
| Mental state:Executive | | | -0.000 |
| | | | (-0.002 – 0.002) |
| Domain:Mental state:Executive | | | 0.001 |
| | | | (-0.002 – 0.005) |
| **Random Effects** | | | |
| $\sigma^2$ | 0.25 | 0.26 | 0.25 |
| $\tau_{00}$ | 5.83 | 5.82 | 5.83 |
| ICC | 0.04 | 0.04 | 0.04 |
| N | 544 $_{ID}$ | 544 $_{ID}$ | 544 $_{ID}$ |
| Observations | 2176 | 2176 | 2176 |
| Marginal $R^2$ / Conditional $R^2$ | 0.494 / 0.536 | 0.494 / 0.536 | 0.494 / 0.535 |

(1) mental state × domain × alerting network; (2) mental state × domain × orienting network; (3) mental state × domain × executive control network. Factors were deviation coded–domain: -.5 harm, .5 purity; mental state: -.5 intentional, .5 accidental. ANT scores were mean-centered. *p < .05; **p < .01; ***p < 0.001. 95% credible intervals are present within brackets.

of innocent intentions in purity (vs. harm) violations. As shown in the top panel of Fig 3, the mental state × domain interaction can be seen at low (executive control score mean +1 SD) and high (executive control score mean -1 SD) levels of executive control network efficiency.

As would be expected, the parameters for the Bayesian model with uninformative priors were very similar (see Table 2). 100% of the posterior distribution for all of the mental state × domain × attentional network score interactions were in the region of practical equivalence, ROPE: [-0.25 0.25] (see S4-S6 Figs in Supplementary Figs at https://osf.io/8k2hj/). Put another way, 100% of the posterior distribution for all of the three-way interactions were $0 \pm .1{*}SD$ or half of a "small effect" ($d = 0.1$). The model without the mental state × domain × alerting network score interaction was 75 times more likely than a model including this term, $BF_{01} = 74.63$. The model without the mental state × domain × orienting network scores interaction was over 200 times more likely than a model including this term, $BF_{01} = 206.74$. Finally, the model without the mental state × domain × executive control network scores was 75 times more likely than a model including this term, $BF_{01} = 75.08$.

Taken together, these results offer little support for attentional failure as an alternative explanation of the reduced exculpatory value of innocent intentions in purity (vs. harm) violations. This is indicated by the very small effect sizes, the small, non-significant coefficients (*b*s. range from .001 to .003) and corresponding small credible intervals, the large Bayes Factors in support of the model without the three-way interaction, and the percentage (100%) of the posterior distribution for the three-way interactions that are in the region of practical equivalence. While the evidence is not consistent with the attention failure explanation of the mental state × domain effect, these results are consistent with the moral pluralism account: the view that separate, domain-specific moral systems explain the reduced exculpatory value of innocent intentions in purity (vs. harm) violations.

## Discussion

This carefully designed and well-powered experiment does not find any evidence that attentional failure explains the reduced exculpatory value of innocent intentions in purity (vs. harm) violations. We postulated that failure to integrate mental state information into the computations underpinning moral cognition for purity violations might be caused by the weirdness (e.g., sexual intercourse with a sibling) of the purity vignettes employed in this literature [9]. Therefore, we reasoned that individuals high (vs. low) in attentional efficiency should show a weaker, or no, mental state × domain effect. However, the efficiency of all three attentional networks (alerting, orienting, and executive control) did not moderate the mental state × domain effect. This finding is consistent with the idea that domain-specific information processing explains the reduced exculpatory value of innocent intentions across domain [1,2]. Put differently, our findings are in line with moral pluralism and do not lend support to the idea that more domain-general processes like attention may explain such regularities in our moral cognition [8,11]. That said, it is possible that attentional processes may still provide an alternative (to separate moral systems) explanation of the mental state × domain effect.

While the ANT is a well-validated measure of attentional processes [14,21,22], individual differences in attentional network efficiency may not be the best candidate for an attention-based explanation of the mental state × domain effect. Another possible avenue might be the causal manipulation of attentional processes through manipulating the saliency of mental state information in the moral vignettes. The idea being that if the mental state × domain effect is driven by the weirdness of the purity vignettes making mental state information less salient, then making mental state information highly salient should "knock out" the mental state × domain effect. That said, recent work shows that ("top-down") instructions to focus on

"why" (vs. how) harm and purity actions are being carried out does not attenuate the mental state × domain effect [23]. However, it is possible that "bottom-up" (e.g., directly manipulating stimulus salience) approaches may prove more effective at manipulating the salience of mental state information than the kind of task instructions employed in [23].

Eye-tracking methods that allow more direct, fine-grained recording of attentional processing might be well suited for capturing brief and potentially unconscious shifts in attention to mental state information across harm and purity moral vignettes [24]. The proportion of attention toward mental state information can act as a specific measure of its weighting in the moral judgment process [24]. Indeed, such methods could even potentially allow for causal manipulation of moral judgments through making decision prompts contingent on eye movements in relation to mental state information [25]. Adapting the paradigm in [25], participants could be asked to re-read moral vignettes until a decision prompt appears. Saliency of mental state information can then be manipulated by displaying decision prompts when gaze is fixed (vs. not fixed) on mental state information. Another added benefit of such methods is that they may also afford testing of manipulation-of-mediator and measurement-of-mediation designs [26], something that makes little theoretical sense for individual difference measures of attentional processes like the ANT [27].

Given theory of mind tasks, like integrating mental state information into moral cognition, require domain-general cognitive processes [15,17,18], there are a range of executive functions that may provide a more domain-general explanation for the mental state × domain effect. The weirdness of purity vignettes may lead to failures in inhibition, working memory and cognitive flexibility [28]. These could all be tested in a similar manner to that employed in present study, employing a wide range of associated tasks [28]. In addition to testing alternative domain-general explanations of the mental state × domain effect, future work should derive some positive predictions from domain-specific explanations. For instance, one might expect judgments of vignettes involving both harm and purity violations to take longer than carefully controlled vignettes involving violations from only one moral domain. Such a reaction time approach has proved a useful way to delineate and decompose cognitive phenomenon [4].

An important limitation of the present study is that it is based on one sole set of stimuli for each domain. As such, the extent to which we can generalise to all harm and purity violations is uncertain. Perhaps attentional efficiency would play a greater role if we sampled other stimuli. Such stimuli sampling issues are an important problem for social cognition research [29], having already undermined some of the classic effects in moral cognition research [30]. Future work would do well to test the attentional failure explanation, and other domain-general accounts, using some of the larger sets of harm and purity vignettes in the literature [6,23]. This could also include the use of more "naturalistic" harm and purity vignettes that have been shown to be more closely matched on weirdness and severity [9]. As well as adding to the generalisability of, and potential boundary conditions on, any test of the attentional process account, inclusion of such stimuli would also allow to further test the generalisability of the mental state × domain effect itself. If the weirdness of purity violations drives the mental state × domain effect, then we would expect the effect to be attenuated when less weird purity stimuli are employed.

This carefully conducted and well-powered study does not provide final or absolute evidence that the mental state × domain effect is not explained by attentional processes but it does provide evidence that individual differences in attentional efficacy seem to play little role in the reduced exculpatory value of innocent intentions in purity (vs. harm) violations. We hope this paper inspires efforts to test domain-general and domain-specific accounts of the mental state × domain effect, so that we might better understand the functional architecture of our moral minds.

## Author Contributions

**Conceptualization:** Joseph Sweetman.

**Data curation:** Joseph Sweetman, George A. Newman.

**Formal analysis:** Joseph Sweetman.

**Funding acquisition:** Joseph Sweetman, George A. Newman.

**Investigation:** Joseph Sweetman, George A. Newman.

**Methodology:** Joseph Sweetman, George A. Newman.

**Project administration:** George A. Newman.

**Software:** George A. Newman.

**Supervision:** Joseph Sweetman.

**Writing – original draft:** Joseph Sweetman.

**Writing – review & editing:** Joseph Sweetman, George A. Newman.

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
