## [Decision Letter · Decision Letter 0]

18 May 2020

PONE-D-20-11124

Attentional efficiency does not explain the mental state × domain effect

PLOS ONE

Dear Dr. Sweetman,

Thank you for submitting your manuscript to PLOS ONE. After careful consideration, we feel that it has merit but does not fully meet PLOS ONE’s publication criteria as it currently stands. Therefore, we invite you to submit a revised version of the manuscript that addresses the points raised during the review process.

Please find below the reviewers' and mine's comments.

We would appreciate receiving your revised manuscript by Jul 02 2020 11:59PM. To enhance the reproducibility of your results, we recommend that if applicable you deposit your laboratory protocols in protocols.io, where a protocol can be assigned its own identifier (DOI) such that it can be cited independently in the future. For instructions see: http://journals.plos.org/plosone/s/submission-guidelines#loc-laboratory-protocols

We look forward to receiving your revised manuscript.

Kind regards,

Valerio Capraro

Academic Editor

PLOS ONE

Journal Requirements:

Additional Editor Comments (if provided):

I have now collected two reviews from two experts in the field. Both reviewers are extremely happy about your paper and have only a few minor comments. I have read the paper myself and I must say that I share their judgment. Therefore, I would like to invite you to revise your work according to the reviewers' comments. Moreover, while reading, I had a thought which I am not sure how relevant is (but it should be). In a paper with Jim Everett and Brian Earp, we also found some evidence of "moral pluralism" using something similar (I think) to a mental state x moral domain effect, although in a different context. As scale we had the Oxford Utilitarianism Scale (Kahane et al., Psych Rev, 2018), which measures how utilitarian a subject is along two dimensions, instrumental harm and impartial beneficence; half of the participants took this scale when primed to follow their emotion and the other half when primed to follow their reason. We found that priming emotion decreases instrumental harm, while leaving impartial beneficence unaffected (Capraro, Everett, & Earp, J Exp Soc Psych, 2019). It seems related to your point (mental state seen as system 1 vs system 2 and moral domain seen as impartial beneficence vs instrumental harm), but I am not super sure. So, feel free to ignore this comment if you feel it is unrelated.

I am looking forward to receive the final version.

Reviewers' comments:

Reviewer's Responses to Questions

**Comments to the Author**

1. Is the manuscript technically sound, and do the data support the conclusions?

Reviewer #1: Yes

Reviewer #2: Yes

2. Has the statistical analysis been performed appropriately and rigorously? 

Reviewer #1: Yes

Reviewer #2: Yes

3. Have the authors made all data underlying the findings in their manuscript fully available?

Reviewer #1: Yes

Reviewer #2: Yes

4. Is the manuscript presented in an intelligible fashion and written in standard English?

Reviewer #1: Yes

Reviewer #2: Yes

5. Review Comments to the Author

Reviewer #1: A number of studies have found that people rely less on intent information when judging purity violations relative to harm violations. This finding has been presented as evidence that purity and harm violations belong to distinct moral domains that are evaluated by different cognitive mechanisms. Sweetman and Newman set out to investigate an alternative explanation for this pattern, hypothesizing that the reduced impact of intent for judgments of purity violations may result from the relative weirdness of purity scenarios, such that weirdness elicits attentional failures that prevent people from processing and integrating intent information. The authors tested this hypothesis by measuring individual differences in attentional efficiency, with the prediction that those high (vs low) in attentional efficiency would be less likely to show reduced sensitivity to intent information when evaluating purity violations. They do not find any moderating effect of attentional efficiency, and conclude that this finding is consistent with moral pluralism.

The authors should be applauded for their well-designed, well-powered work that uses sophisticated analysis techniques and open science approaches (that represent improvements on the past relevant work). I have only a few minor comments for their consideration.

1) On page 4, the authors write that “convergent evidence has been found both in fMRI and cross-cultural work”. I recommend they describe the convergent evidence from these studies in some detail.

2) The authors write that “the weirdness of purity (vs. harm) scenarios may compete with standard moral cognition for attentional resources” (p5). Has this general effect of weirdness on attentional resources ever been tested directly in prior studies? If so, the authors should cite studies that provide evidence of this relationship.

3) The authors speculate on p15 that “if the mental state × domain effect is driven by the weirdness of the purity vignettes making mental state information less salient, then making mental state information highly salient should “knock out” the mental state × domain effect.” This hypothesis was recently tested in an fMRI study: Dungan & Young (2019), SCAN. The authors found that prompting participants to focus on intent information did not knock out the mental state x domain effect.

4) The section on page 15, beginning with “Indeed, such methods…” and ending with “measure of attentional processes like the ANTs”, is a bit unclear. It would help if the authors could explain what it would mean to make decision prompts contingent on eye movements in the context of a moral judgment task.

Typos:

1) Cue is mis-spelled que in several places.

2) There are a few grammatical errors in this sentence:

Another added benefit of such methods is that they may also afford testing of manipulation-of-mediator and measurement-of-mediation designs [23], something that makes little theoretically sense for individual difference measure of attentional processes like the ANT [24].

Reviewer #2: The question of whether moral judgment is domain-specific (moral pluralism) or domain general is an important one, and we commend the authors for taking a novel approach to this long-standing debate, with the specific focus on intent and attentional efficiency. Overall, the article is well written and engaging. Most of our comments focus on strengthening the Introduction. We offer the following feedback:

1. In the early parts of the introduction I would have appreciated seeing a more thorough definition of what is meant by ‘harm’ versus ‘purity’ violations. Examples are provided, but not explanations of the particular characteristics of each domain. As a result, readers who are not familiar with the moral foundations might see the narrow examples provided in the study as fully representative of the relevant domains. As your paper has one study and is thus not too long, I would like to see a more comprehensive introduction including the specific points in the following comments. PLOS readers tend to be multi-disciplinary and will need more fleshed out introductions to these areas of research.

2. On page 3, the examples provided for each “domain of action” are all violations – not all moral actions are ones that violate, so I would suggest introducing your examples as violations or providing examples that represent virtue and vice.

3. The introduction should also include a brief section on discrete moral emotions, an area that has contributed to the moral pluralism debate, see Cameron, Lindquist, & Gray (2015) in PSPR and the studies they review in their paper

4. When introducing moral pluralism, include a brief discussion of the critiques – mainly Kurt Gray and colleagues’ work that everything essentially boils down to harm – even purity violations.

5. On page 4 under the ‘Moral systems or attentional failures’ heading the authors introduce the central argument for an alternative explanation to domain-specific moral systems, namely, that purity scenarios depict violations that are “weirder” and less severe than cases of harm. Again, I would have preferred to see more clarifying detail on what is meant by ‘weirdness’ and ‘severity’ and further explanation of how cited authors – and the authors of the manuscript – claim that this influences intention effects.

6. On page 4, change vs, to v. at bottom of page

7. p. 6. The relationship between attentional control and integration of mental state information is presented as a ‘given’, however I would have like this to be explained or justified further. What reasons are there to think that integration of mental state information is affected by attention control in ways that other aspects of moral cognition are not?

8. In the ANT section of Materials and measures, the authors have used two different spellings of ‘cue’.

9. I wonder whether the selection of the incest scenario for the purity violation may have confounded the intention-accident distinction. People feel so strongly disgusted by incest, and this scenario has probably been labelled as one of the weirdest in the purity set – perhaps less so than the ones about eating your dead dog. Can you include something to justify how the incest and poisoning vignettes are matched.

10. Regarding the coding error and the failure to record participant gender and age (need a comma after coding error in the footnote), can you include a note to dismiss any concerns – assuming that you did not expect to find any individual differences based on gender or age

11. I could not find where the coding for mental state or domain was recorded, making it difficult to interpret the direction of effects in the models. Similarly, descriptive statistics do not seem to have been reported.

12. Given that the central premise of the paper is that the ‘weirdness’ of purity violations interferes with attention that may otherwise be paid to intentionality I would have liked this ‘weirdness’ to be investigated as a variable independent of domain. As Gray and Keeney (2015) have shown, it is possible to conceive of purity violations that are not ‘weird’ and harm violations that are. Without such investigation I feel it is difficult to generalise the intentionality effect equally to all purity violations. This should be mentioned in your limitations and areas for future research.

13. The authors base some of their theorising on Huebner, Dwyer, & Hauser’s (2009) finding that emotion may impact moral judgments through its impact on attention processes. As an alternate explanation for their results, have the authors considered the possibility that both the heightened emotional power of the harm vignettes and the weirdness of the purity vignettes similarly co-opt cognitive processing?

6. PLOS authors have the option to publish the peer review history of their article (what does this mean?). If published, this will include your full peer review and any attached files.

Reviewer #1: No

Reviewer #2: Yes: Dr Melissa A. Wheeler and Ms Melanie J. McGrath

---

## [Author Response · Author response to Decision Letter 0]

25 May 2020

See attached "Response to reviewers" also copy and pasted below:

Response to reviewers (PONE-D-20-11124)

Editor comments

Moreover, while reading, I had a thought which I am not sure how relevant is (but it should be). In a paper with Jim Everett and Brian Earp, we also found some evidence of "moral pluralism" using something similar (I think) to a mental state x moral domain effect, although in a different context. As scale we had the Oxford Utilitarianism Scale (Kahane et al., Psych Rev, 2018), which measures how utilitarian a subject is along two dimensions, instrumental harm and impartial beneficence; half of the participants took this scale when primed to follow their emotion and the other half when primed to follow their reason. We found that priming emotion decreases instrumental harm, while leaving impartial beneficence unaffected (Capraro, Everett, & Earp, J Exp Soc Psych, 2019). It seems related to your point (mental state seen as system 1 vs system 2 and moral domain seen as impartial beneficence vs instrumental harm), but I am not super sure. So, feel free to ignore this comment if you feel it is unrelated.

Response: Thanks for letting us know about this interesting work. There are some peripheral similarities (different effects across responses to different types of moral stimuli) with our work. However, it is unclear to me just where “impartial beneficence” fits into popular accounts of moral pluralism (e.g., moral foundations theory). Having looked at your paper it seems that it is a mix of justice and harm concerns. Looking at responses to stimuli that represent multiple moral domains is an interesting general avenue for future work, as is examining how utilitarianism and dual process accounts map on to moral pluralism, but we feel this is beyond the scope of the present paper. Here we have a specific focus on explanations of the classic mental state x domain effect. 

Reviewer 1

This reviewer raised four issues:

Issue 1: On page 4, the authors write that “convergent evidence has been found both in fMRI and cross-cultural work”. I recommend they describe the convergent evidence from these studies in some detail.

Response: We have now included additional details of the converging evidence for the mental state × domain interaction (p.4).

Issue 2: The authors write that “the weirdness of purity (vs. harm) scenarios may compete with standard moral cognition for attentional resources” (p5). Has this general effect of weirdness on attentional resources ever been tested directly in prior studies? If so, the authors should cite studies that provide evidence of this relationship.

Response: To our knowledge, this has not been directly tested in terms of “weirdness” but a fair amount of work has shown that surprising and unexpected stimuli, which we argue purity violations like incest and drinking urine are clear cases of, compete for attentional resources and can lead to “surprise-induced blindness” (analogous to attentional blink) in attentional processes. We have now added details of this (p5).

Issue 3: The authors speculate on p15 that “if the mental state × domain effect is driven by the weirdness of the purity vignettes making mental state information less salient, then making mental state information highly salient should “knock out” the mental state × domain effect.” This hypothesis was recently tested in an fMRI study: Dungan & Young (2019), SCAN. The authors found that prompting participants to focus on intent information did not knock out the mental state x domain effect.

Response: We have now added discussion of this study (p15). Specifically, we suggest that more “bottom-up” (e.g., directly manipulating stimulus salience) approaches could prove more effective than “top-down” manipulation of task goals for manipulating the salience of mental state information.

Issue 4: The section on page 15, beginning with “Indeed, such methods…” and ending with “measure of attentional processes like the ANTs”, is a bit unclear. It would help if the authors could explain what it would mean to make decision prompts contingent on eye movements in the context of a moral judgment task.

Response: We have briefly sketched out how one might do this (p16-17).

This reviewer also was kind enough to point out some typos/grammatical errors:

Typos:

1) Cue is mis-spelled que in several places.

2) There are a few grammatical errors in this sentence:

Another added benefit of such methods is that they may also afford testing of manipulation-of-mediator and measurement-of-mediation designs [23], something that makes little theoretically sense for individual difference measure of attentional processes like the ANT [24].

Response: We have corrected these typos and grammatical errors and are grateful to the reviewer for spotting them.

Reviewers 2a and 2b

These reviewers raised over a dozen issues. We have gone through these with careful thought and acted on them where we believed that it would make for a better manuscript:

Issue 1: In the early parts of the introduction I would have appreciated seeing a more thorough definition of what is meant by ‘harm’ versus ‘purity’ violations. Examples are provided, but not explanations of the particular characteristics of each domain. As a result, readers who are not familiar with the moral foundations might see the narrow examples provided in the study as fully representative of the relevant domains. As your paper has one study and is thus not too long, I would like to see a more comprehensive introduction including the specific points in the following comments. PLOS readers tend to be multi-disciplinary and will need more fleshed out introductions to these areas of research.

Issue 2: On page 3, the examples provided for each “domain of action” are all violations – not all moral actions are ones that violate, so I would suggest introducing your examples as violations or providing examples that represent virtue and vice.

Response: We have added more details on this (p3), but we have balanced this with keeping our paper short and focused on testing attentional explanations of the mental state x domain effect. There are ample citations to moral foundations theory in the paper, including lengthy review articles which interested readers can consult if they are interested in the considerable scope of moral foundations theory. 

Issue 3: The introduction should also include a brief section on discrete moral emotions, an area that has contributed to the moral pluralism debate, see Cameron, Lindquist, & Gray (2015) in PSPR and the studies they review in their paper.

Issue 4: When introducing moral pluralism, include a brief discussion of the critiques – mainly Kurt Gray and colleagues’ work that everything essentially boils down to harm – even purity violations.

Response: Examining the role of emotion and the degree to which harm-based “moral templates” can explain evidence for moral pluralism is an interesting topic but given that emotion and Gray’s “harm pluralism” played little role in our theorising or our empirical work we think that reviewing this work will only detract from reporting a theoretically focused and carefully conducted study. To be clear, we tested whether attentional failure could account for the mental state x domain effect. If we had (counterfactually) found that it did, this would not support the idea that our moral cognition for purity violations is based on a harm-based moral template. It would only suggest that key evidence for moral pluralism (i.e., separate, domain-specific moral systems) is better explained by an attentional account. Again, introducing emotion and Gray’s harm pluralism into the introduction of a paper not testing emotion or Gray’s harm-based account, or suggesting any future directions or implications for these approaches seems to take us beyond the scope of our paper and only detracts from the key theoretical contribution of the paper: attentional failure does not explain the mental state x domain effect.

Issue 5: On page 4 under the ‘Moral systems or attentional failures’ heading the authors introduce the central argument for an alternative explanation to domain-specific moral systems, namely, that purity scenarios depict violations that are “weirder” and less severe than cases of harm. Again, I would have preferred to see more clarifying detail on what is meant by ‘weirdness’ and ‘severity’ and further explanation of how cited authors – and the authors of the manuscript – claim that this influences intention effects.

Response: We’ve added extra detail on weirdness and severity (p4). In addition, we have added extra details of how weird or unexpected stimuli (like purity violations) may compete for attentional resources, thus making mental state information less salient (see response to Reviewer 1, issues 2; p5).

Issue 6: On page 4, change vs, to v. at bottom of page 

Response: We’ve corrected this typo and thank the reviewers for spotting it.

Issues 7: p. 6. The relationship between attentional control and integration of mental state information is presented as a ‘given’, however I would have like this to be explained or justified further. What reasons are there to think that integration of mental state information is affected by attention control in ways that other aspects of moral cognition are not?

Response: We provide additional citations (p. 7) for work in mental state reasoning/theory of mind (ToM) that demonstrates that attentional control is employed in theory of mind tasks that involve integrating mental state information with other types of information. This is widely excepted in the ToM literature, forming a large part of discussions as to which tasks use “purer” measures of ToM. We do not provide reasons for thinking attentional control plays no role in other aspects of moral cognition because we do not think that is likely to be the case. The only reason we focus on the possible role of attentional control in mental state information is because we are trying to test possible attentional explanations for an effect that is based on differential use of mental state information across moral domains. We think this is now clearer with this and other changes that we have made.

Issue 8: In the ANT section of Materials and measures, the authors have used two different spellings of ‘cue’.

Response: We’ve corrected this typo and thank the reviewers for spotting it (see Reviewer 1 Typos).

Issue 9: I wonder whether the selection of the incest scenario for the purity violation may have confounded the intention-accident distinction. People feel so strongly disgusted by incest, and this scenario has probably been labelled as one of the weirdest in the purity set – perhaps less so than the ones about eating your dead dog. Can you include something to justify how the incest and poisoning vignettes are matched.

Response: The original Young & Saxe (2011) paper that employed this pair of scenarios empirically examined (Experiment 2) whether the difference in moral judgments of harm versus purity violations was due to differences in the emotional salience of harm vs. purity (i.e., incest) scenarios or the agent’s mental state (intentional vs. accidental). They found no support for these potential confounds. It’s beyond the scope of our paper to address confounds that have already been addressed some 10 years ago. We’d suggest publishing the review history for those that might be thinking the same as the reviewers. This is a good way of allowing a paper to remain concise and focused, without having to address thoughts that have been previously addressed in earlier work. 

Issue 10: Regarding the coding error and the failure to record participant gender and age (need a comma after coding error in the footnote), can you include a note to dismiss any concerns – assuming that you did not expect to find any individual differences based on gender or age.

Response: We have added to the footnote to address this.

Issue 11: I could not find where the coding for mental state or domain was recorded, making it difficult to interpret the direction of effects in the models. Similarly, descriptive statistics do not seem to have been reported.

Response: We are grateful to the reviewers for spotting this and have added full details of the coding to the reformatted regression tables (p 11, 14). We have also added details of the descriptives in a “Supplementary Descriptives” file on the osf page and have directed readers to this (p11). Of course all the data is available of the osf page so anybody can examine, explore, and test any ideas they may have when reading the paper – on of the advantages of open science!

Issue 12: Given that the central premise of the paper is that the ‘weirdness’ of purity violations interferes with attention that may otherwise be paid to intentionality I would have liked this ‘weirdness’ to be investigated as a variable independent of domain. As Gray and Keeney (2015) have shown, it is possible to conceive of purity violations that are not ‘weird’ and harm violations that are. Without such investigation I feel it is difficult to generalise the intentionality effect equally to all purity violations. This should be mentioned in your limitations and areas for future research.

Response: We have addressed this in our discussion of limitations and future directions (p17).

Issue 13: The authors base some of their theorising on Huebner, Dwyer, & Hauser’s (2009) finding that emotion may impact moral judgments through its impact on attention processes. As an alternate explanation for their results, have the authors considered the possibility that both the heightened emotional power of the harm vignettes and the weirdness of the purity vignettes similarly co-opt cognitive processing?

Response: No, we have not considered this and are not exactly sure what it would mean. If we understand the reviewers correctly, then it would seem to suggest that the emotional saliency of harmful acts and the weirdness of purity acts should both lead to attentional failures. We have empirical reasons to doubt the former as we detailed in our response to issue 9: “The original Young & Saxe (2011) paper that employed this pair of scenarios empirically examined (Experiment 2) whether the difference in moral judgments of harm versus purity violations was due to differences in the emotional salience of harm vs. purity (i.e., incest) scenarios or the agent’s mental state (intentional vs. accidental).” Indeed, weirdness is the only potential confound that we have empirical reasons to think just might drive any attentional effect, which is why we chose to focus on it and its potential impact on attentional processing.

---

## [Editor Report · Decision Letter 1]

28 May 2020

Attentional efficiency does not explain the mental state × domain effect

PONE-D-20-11124R1

Dear Dr. Sweetman,

We are pleased to inform you that your manuscript has been judged scientifically suitable for publication and will be formally accepted for publication once it complies with all outstanding technical requirements.

With kind regards,

Valerio Capraro

Academic Editor

PLOS ONE
---

## [Editor Report · Acceptance letter]

1 Jun 2020

PONE-D-20-11124R1 

Attentional efficiency does not explain the mental state × domain effect 

Dear Dr. Sweetman:

I am pleased to inform you that your manuscript has been deemed suitable for publication in PLOS ONE. Congratulations! Your manuscript is now with our production department. 

With kind regards,

on behalf of

Dr. Valerio Capraro 

Academic Editor

PLOS ONE